# Impact and Effectiveness of COVID-19 Vaccines Based on Machine Learning Analysis of a Time Series: A Population-Based Study

**DOI:** 10.3390/jcm13195890

**Published:** 2024-10-02

**Authors:** Rafael Garcia-Carretero, Maria Ordoñez-Garcia, Oscar Vazquez-Gomez, Belen Rodriguez-Maya, Ruth Gil-Prieto, Angel Gil-de-Miguel

**Affiliations:** 1Internal Medicine Department, Mostoles University Hospital, Rey Juan Carlos University, 29835 Mostoles, Spain; govaz55@gmail.com (O.V.-G.); belenrmaya@gmail.com (B.R.-M.); 2Hematology Department, Mostoles University Hospital, Rey Juan Carlos University, 29835 Mostoles, Spain; 3Department of Preventive Medicine and Public Health, Rey Juan Carlos University, 28933 Madrid, Spain; ruth.gil@urjc.es (R.G.-P.); angel.gil@urjc.es (A.G.-d.-M.)

**Keywords:** COVID-19, vaccines, SARS-CoV-2, hospitalizations, mortality, machine learning

## Abstract

**Background**: Although confirmed cases of infection with severe acute respiratory syndrome coronavirus 2 (SARS-CoV-2) have been declining since late 2020 due to general vaccination, little research has been performed regarding the impact of vaccines against SARS-CoV-2 in Spain in terms of hospitalizations and deaths. **Objective**: Our aim was to identify the reduction in severity and mortality of coronavirus disease 2019 (COVID-19) at a nationwide level due to vaccination. **Methods**: We designed a retrospective, population-based study to define waves of infection and to describe the characteristics of the hospitalized population. We also studied the rollout of vaccination and its relationship with the decline in hospitalizations and deaths. Finally, we developed two mathematical models to estimate non-vaccination scenarios using machine learning modeling (with the ElasticNet and RandomForest algorithms). The vaccination and non-vaccination scenarios were eventually compared to estimate the number of averted hospitalizations and deaths. **Results**: In total, 498,789 patients were included, with a global mortality of 14.3%. We identified six waves or epidemic outbreaks during the observed period. We established a strong relationship between the beginning of vaccination and the decline in both hospitalizations and deaths due to COVID-19 in all age groups. We also estimated that vaccination prevented 170,959 hospitalizations (CI 95% 77,844–264,075) and 24,546 deaths (CI 95% 2548–46,543) in Spain between March 2021 and December 2021. We estimated a global reduction of 9.19% in total deaths during the first year of COVID-19 vaccination. **Conclusions**: Demographic and clinical profiles changed over the first months of the pandemic. In Spain, patients over 80 years old and other age groups obtained clinical benefit from early vaccination. The severity of COVID-19, in terms of hospitalizations and deaths, decreased due to vaccination. Our use of machine learning models provided a detailed estimation of the averted burden of the pandemic, demonstrating the effectiveness of vaccination at a population-wide level.

## 1. Introduction

The coronavirus disease 2019 (COVID-19) pandemic has had a significant impact on the health of the population, as well as significant implications in all sectors of society and the daily lives of citizens [1,2,3,4]. It is claimed that high levels of vaccination coverage, the characteristics of the omicron variant, and increased diagnostic testing likely contributed to the observed impact of the pandemic in the last months of 2021. In addition, there was a very high incidence of confirmed cases, but a majority of these had mild symptoms or were asymptomatic. This placed a significant strain on primary health care rather than hospitals. Therefore, the occupancy percentage of hospital and intensive care unit (ICU) beds was much lower than expected relative to what occurred over the remainder of the pandemic [5,6,7,8].

By February 2022, more than 92% of the Spanish population over the age of 12 was fully vaccinated [9]. Current evidence indicates that the various COVID-19 vaccines have achieved high levels of effectiveness in restricting moderate and severe forms of the disease and reducing lethality. Vaccines, despite reducing the probability of infection, are less effective at completely preventing virus replication in the upper respiratory mucosa of a vaccinated individual, which means that transmission is possible from vaccinated individuals who have been infected, even if the disease is mild or asymptomatic [10,11,12,13,14,15]. This makes it infeasible to aim for the virus’s eradication at present. Therefore, researchers should focus their efforts toward reducing the severity of infections while maintaining a level of transmission that is manageable and does not generate an excessive burden on the healthcare system.

As noted, due to the increase in vaccination coverage and the immunity generated from natural infections, the majority of the population is protected against severe COVID-19 [16]. Data show that protection has been maintained, even against a variant antigenically different enough from the previous ones to produce very high incidence rates in the population that previously had immunity.

Observational studies, such as case–control or cohort studies, are not always feasible, so several studies using alternative approaches have been conducted to demonstrate the effectiveness of vaccination [8,17]. Likewise, several meta-analyses have studied the effectiveness of vaccination from the following three perspectives: efficacy against infection, efficacy against severe disease (and, hence, reduction in risk of hospital admission), and ability to reduce the transmissibility of vaccinated individuals who become infected [13,15,16,18]. However, the impact of vaccination in terms of decreasing hospitalizations and deaths has not yet been investigated in a nationwide, population-based, epidemiological study in Spain.

Given the unique characteristics of the Spanish healthcare system and the country’s age-stratified vaccination strategy, studying Spain offers an opportunity to understand the differential impact of vaccination across diverse demographic groups, contributing insights that are not directly generalizable to other populations.

### Hypothesis and Objectives of Our Research

We designed a population-based study to assess vaccination as a major public health intervention. By this means, we investigated whether vaccines have been beneficial to the Spanish population. Our research objective was to determine whether vaccination reduced the number of hospitalizations and deaths. We conducted our study in three stages. First, we described the differences between two periods, namely the first months of the pandemic, during which no vaccination was present, and the last months of 2021, when a high proportion of the Spanish population was vaccinated. Secondly, we compared trends in hospitalizations and deaths with the vaccination rate. Finally, we assessed the effectiveness of vaccines against severe disease in terms of the reduction in hospitalizations and mortality due to COVID-19. We estimated the number of averted hospitalizations and deaths. We also compared the evolution of the pandemic across the following two scenarios: vaccination (the observed scenario) and non-vaccination (an estimated scenario). The estimated scenario was fitted using time series and machine learning analyses.

## 2. Materials and Methods

### 2.1. Data Collection and Study Design

We designed a retrospective, population-based study using data collected from electronic health records. We collected data from the Spanish Minimum Basic Data Set at Hospitalization (MBDS-H), provided by the Spanish Ministry of Health [19]. We also collected data related to COVID-19 vaccination in the European Union/European Economic Area (EU/EEA) from the European Centre for Disease Prevention and Control [20]. Figure A1 shows a flow chart of the study.

MBDS-H is a mandatory administrative registry of hospital discharges that covers more than 95% of Spanish hospitals, including public centers in the National Spanish Health System and private hospitals. Nearly 97% of total hospital discharges are covered in the database. The MBDS-H is exclusively built from discharge reports. Microdata from patients include information on sex, age, dates of admission and discharge, type of discharge, primary and secondary diagnoses at discharge, length of stay, and surgical or obstetric procedures, among other data. Other administrative data are recorded by default, including the province where the hospitalization occurred, place of residence, and cost of hospitalization. By default, the Ministry of Health provides de-identified data to ensure patient privacy; thus, no names or personal information were recorded. The purpose of the MBDS-H is to facilitate the development of retrospective studies for the calculation of the burden of hospitalization and assessment of risk factors from thousands of patients, i.e., enabling population-based studies. From 2016 onward, MBDS-H has used the coding system of the International Classification of Diseases, 10th edition. MBDS-H is considered a valuable system for the epidemiological analysis of any coded disease.

Vaccination data were downloaded from the European Centre for Disease Prevention and Control [20]. These data were collected through the European Surveillance System. All EU/EEA Member States are requested to report basic indicators on vaccination (vaccines categorized by manufacturer, number of doses administered, vaccinated population, etc.). Data are categorized by target and age group at a national level.

### 2.2. Inclusion and Exclusion Criteria

In this retrospective study, cases were collected from the MBDS-H from the Spanish Ministry of Health. We included all patients with the code for COVID-19 (U07.1) in any diagnostic position (either primary or secondary diagnosis) from 1 January 2020 to 31 December 2021.

All age groups were studied, with special emphasis on those older than 60 years of age. We analyzed the healthcare impacts in terms of mortality and ICU admission by dividing the population into age groups. Patients with incomplete data regarding ICU admission, mortality, length of stay, or COVID-19 disease were excluded. We excluded patients with unknown data to ensure the accuracy and completeness of the dataset. As length of stay is a key outcome variable in the analysis of disease severity and healthcare utilization, including patients with missing values could bias the results and reduce the robustness of our conclusions.

### 2.3. Definition of Waves

We categorized the pandemic following a previous epidemiological study [21]. Using only data from Spain, we split the entire pandemic period into outbreaks or epidemiological waves based on the 14-day cumulative incidence, which marked the turning point for each wave. Every turning point indicated the end of one wave and the beginning of the next one, similar to the methodology used in previous epidemiological studies [22].

As mentioned in the introduction, the first and second stages of our study were descriptive. We analyzed the evolution of the pandemic and its outbreaks, comparing the first waves, during which time vaccination was absent, with the last waves of 2021, when vaccination was present. Herein, we describe the demographic and epidemiological differences between the two periods and their relationships with vaccination.

### 2.4. Vaccination Rollout

The Vaccination Strategy Against COVID-19 in Spain was developed by the Spanish Ministry of Health [23]. The working group prioritized certain age groups to receive the vaccine based on the supply of doses and the availability of current evidence, taking into consideration the demographic characteristics of the Spanish population [24]. Assessments of ethical concerns and risk factors were also considered to prioritize certain age groups over others. The elderly and healthcare workers were the first groups to receive the vaccine, and the rollout moved forward through the rest of the age groups over the course of 2021. We assessed the trends of vaccination over time using data from the European Centre for Disease Prevention and Control.

We split the population into age groups to assess the evolution of the pandemic in terms of hospitalizations and deaths. Then, we compared the vaccination rates to those trends by age group.

### 2.5. Estimated Scenarios of the Unvaccinated Population

We developed a population-based, epidemiological study to compare the following two scenarios: observed hospitalizations and deaths before and after vaccination and an estimated non-vaccination scenario using time series and machine learning models.

### 2.6. Mathematical Modeling of Hospitalizations and Mortality

We utilized ElasticNet and random forest models to forecast the impact of vaccination by fitting the models to a training dataset from July 2020 to February 2021 and validated them using a testing set. Each model was evaluated using cross-validation metrics (see Appendix A).

### 2.7. Machine Learning Algorithms

The models included ElasticNet for linear predictions and random forest to capture nonlinear patterns. Each algorithm was tuned to achieve optimal performance. The former assumes linearity, and the latter makes no assumptions on linearity. Researchers and data scientists apply machine learning algorithms in various fields, including health care, finance, and natural language processing [25,26,27].

### 2.8. Fitting the Models

To fit the models, we first split our time-series dataset into a training and a testing set. The training set covers the period between 1 July 2020 and 29 February 2021. We excluded the first wave (March to June 2020) because we considered it an outlier that could add noise to the final model. The testing set was not used to develop the models but for comparison purposes only. We made no assumptions on the likelihood, normality, or linearity of the training set. We fit the models to the training set, tuning the hyperparameters of each algorithm to achieve the best accuracy. For EN, we set alpha, and for RF, we set mtry and the numbers of trees. A key mathematical condition when tuning the models was that they should fit accurately with the observed data, i.e., the training dataset.

We used R package *randomforest* for the RF model and R package *glmnet* for the EN model. We fitted the two models to time series of both hospitalizations and deaths. Thus, we computed four models. Once they were developed, we forecasted data for the next months, namely 1 March 2021 to 31 December 2021. Finally, by comparing the estimated hospitalizations and deaths (had vaccination never been implemented) with the observed data, we could explore the number of events that were averted in the last 10 months of 2021 due to vaccination.

### 2.9. Statistical Analyses

All statistical and machine learning-based analyses were conducted using R language version 4.3.2 (Vienna, Austria) [28]. Statistical significance was set at 0.05.

## 3. Results

### 3.1. Nationwide Overview of the Pandemic

We included data from 498,789 hospital admissions (Table 1) and excluded 113 patients due to inconsistent or incomplete data. We split the observation period into waves, as described above (Figure 1). The first waves included more than 115,000 hospitalizations, and this number dropped up to 50,000 in the fourth and fifth waves. Men were admitted at higher rates than women (56.1%, *p* = 0.001), with no changes in the distribution during the pandemic. The median age was 66, but this tended to decrease across the fourth and fifth waves (59 and 57, respectively). Length of stay in both the standard hospitalization ward and in the ICU was more heterogeneous, and no clear trend could be established. Although the nationwide mortality ratio was 14.3% in Spain, we observed a decreasing trend from the first wave (18.2%) to the fourth and fifth waves (7.4% and 10.1%, respectively). Comorbidities such as type 2 diabetes, hypertension, coronary disease, dementia, kidney disease, malignancy (either solid tumor or hematological malignancy), and chronic respiratory disease showed a decreasing trend starting with the fourth wave. Other comorbidities, such as liver and cerebrovascular diseases, showed no changes. Obesity and heart failure showed a more heterogeneous trend.

Table 2 shows disaggregated data of hospital admissions, ICU admissions, and mortality by age group. These data are also represented in Figure A2.

Among hospitalizations, the predominant age group were >60 years old in the second and third waves. The >80-year-old age group dropped dramatically in the fourth and fifth waves, and the group of 18- to 49-year-olds was predominant in the fifth wave. Regarding mortality, deaths in all ages dropped quite evenly, although the patients who were more affected were >60 years old (Figure 2). Figure 2 displays data beginning with the second wave, as details of the following waves are of interest to compare the second and third waves on one side with the fourth and fifth waves on the other.

### 3.2. Vaccination Rollout

Figure 3 plots the vaccination rollout in Spain, both globally and by age group. Vaccination began in December 2020 with the elderly and healthcare workers. By 31 December 2021, 80.3% of the whole Spanish population was fully vaccinated, i.e., having received the complete regimen, and 97.2% had received at least one dose. By April 2021, 75% of >80-year-olds and 48% of >60-year-olds were fully vaccinated. Figure A3 provides more detail on age groups regarding vaccination coverage over time.

### 3.3. Hospitalizations and Deaths in an Estimated Scenario

As noted, our approach involved estimating both hospital admissions and mortality by parsing the time series using machine learning algorithms. We developed four models—one for hospitalizations and another for deaths—using each algorithm. Figure 4 shows the observed and estimated scenarios. Figure A4 shows the models with confidence intervals. Table 3 shows the estimates of hospitalizations and deaths in the absence of vaccination. Using the RF model, we estimated that 251,830 hospitalizations and 37,673 deaths would have occurred in a non-vaccination scenario during the period between March and December 2021. According to the EN model, the estimated numbers of hospitalizations and deaths were 307,617 and 37,141, respectively. Compared to the observed data, we estimated that vaccination prevented 115,172 hospitalizations and 25,078 deaths with the RF model and 170,959 hospitalizations and 24,546 deaths with the EN model. Finally, we plotted Figure 5, showing the cumulative hospitalizations and deaths, with both the RF and EN models.

## 4. Discussion

### 4.1. Descriptive Analyses

We have described the high number of hospitalizations and deaths during the first waves of the pandemic in Spain. We have demonstrated how this trend began to decrease in March to April 2021 as a result of vaccination, which was the major public health intervention during the COVID-19 pandemic.

Overall, the first wave showed the highest number of hospitalizations, the highest mortality rate, the longest hospital and ICU lengths of stay, and the oldest patients. The fourth and the fifth waves showed a decreasing trend in terms of hospitalizations and mortality. In addition, these last waves showed an overall younger, healthier population.

While the sixth wave was included in our analyses, the results might not be reliable, as this wave ended mid-February 2022, and its results are not fully represented in tables and figures. However, the fourth and fifth waves showed that the demographic profile of hospitalized individuals changed with respect to the previous waves, showing a turning point in the evolution of the pandemic.

With respect to admissions by age group, the group of patients under 17 contributed only marginally during the observed period of the pandemic. However, patients over 60 years old were the largest group of those admitted to the hospital due to COVID-19. Patients between 18 and 59 years old were hospitalized at a lower rate. Additionally, most of the deaths occurred in patients >60 years old, particularly in patients over 80 years old, whereas mortality in the rest of the age groups was marginal, as seen in Figure 3.

### 4.2. Vaccination Rollout

Vaccination in Spain began in late December 2020, as soon as vaccines were proven to be safe and to offer significant protection against severe forms of COVID-19, as part of a European initiative [29]. Within only a few weeks of the beginning of vaccination (2.2% of total population by March and 10% by May 2021), we observed a rapid decline in both hospitalizations and deaths beginning in March and April 2021. We also observed a strong temporal correlation between decreasing hospitalizations and deaths on one side and the evolving vaccination rollout on the other (Figure 2 and Figure 3). The decline in hospitalizations and deaths was first observed in patients over 80 years old, showing a relationship between vaccination and protection against both outcomes. This relationship can also be seen in the remaining age groups after the beginning of vaccination. This steady pace of vaccination consolidated the decline in the severity of the pandemic. Our data are also in line with other studies that have investigated the benefit of vaccination and its protective effects [30,31,32]. We can state that in Spain, vaccination led to a significant reduction in the severity of COVID-19 across all age groups, with particularly marked benefits observed in the elderly population. While the overall reduction in hospitalizations and deaths due to vaccination is consistent with global findings, the timing and magnitude of these changes in Spain were influenced by the country’s specific vaccination rollout strategy and healthcare infrastructure.

### 4.3. Modeling and Estimating Data in a Non-Vaccination Scenario

It can be challenging to quantify the impact of vaccination if an incomplete picture of the pandemic is obtained. Infections and confirmed cases are either often under-reported or underestimated [21,33]. For this reason, we relied on reported hospitalizations and deaths to determine this impact instead of trends of non-hospitalized, confirmed cases.

Our reference publication was that of Barandalla et al. [34], who developed simulated curves of hospitalization in the absence of vaccines and compared those curves with the observed incidence. That study investigated hospitalizations in Spain between February 2020 and June 2021. The authors estimated the expected hospitalizations during 2021 in the absence of vaccination, extrapolating data from the second wave. The scenario of an unvaccinated population was estimated to create a statistical model as follows. The authors disaggregated the entire population curve across age groups and took the proportion of hospitalization of age groups of unvaccinated or less vaccinated population as a reference. These proportions of hospitalizations were extrapolated to the remaining groups, yielding curves of the real incidence of hospitalization and curves of expected hospitalization in the absence of vaccines for each age group. Finally, these curves were compared. Showing a decrease in incidence, they demonstrated the beneficial impact of vaccination on hospitalizations. Likewise, vaccine effectiveness against hospitalization in ≥65-year-old age groups was estimated from October 2021 to March 2022 in a European study [35]. The reference group was the unvaccinated population. The authors performed a survival analysis using the Cox proportional hazards regression model to estimate the hazard ratios of hospitalization.

It is beyond the scope of this manuscript to discuss all studies that have used mathematical models to estimate mortality in the absence of vaccination, but it is worth mentioning some of them. A mathematical model reported by Watson et al. [17] estimated that 14.4 million deaths were prevented in 185 countries in 2021. The authors used a framework based on a “susceptible–exposed–infections–recovered–susceptible” model to estimate a non-vaccination scenario. This model was fitted using MCMC, and the authors calculated the time-varying reproductive number to determine the estimated number of contagions. Havers et al. [8] conducted a cross-sectional study that included adults hospitalized with COVID-19, comparing vaccinated versus unvaccinated individuals. Both studies demonstrated the effectiveness of vaccination and its impact on the evolution of the COVID-19 pandemic using different mathematical approaches. Auto-regressive time-series modeling was assessed in other studies [36,37].

In summary, previous studies conducted in Spain [34] have reported a significant reduction in hospitalizations following vaccination rollout using different modeling approaches. Our study adds to these findings by incorporating machine learning methods and providing a more granular age-stratified analysis, which is lacking in other reports. International studies, such as that by Watson et al. [17], have demonstrated similar impacts of vaccination on a nationwide scale, supporting the observed trends in our Spanish cohort.

Machine learning has also been used to estimate the evolution of the COVID-19 pandemic in terms of confirmed cases. Kırbaş et al. [38] conducted a comparative study using different approaches, including ARIMA, neural networks, and long short-term memory (LSTM), to forecast the evolution of the pandemic. LSTM provided predictions with the best accuracy. Neural networks were used by Nabi et al. to study the dynamics of confirmed cases of COVID-19 [39]. Although deep learning (i.e., neural networks) seems to have better prediction accuracy than standard statistical methods (ARIMA) or machine learning, it entails high costs in terms of computational resources and time [40].

Having discussed some of the more relevant publications in this field, we consider this study to stand out due to the use of advanced machine learning algorithms such as ElasticNet and random forest, which allowed us to create accurate non-vaccination scenarios. This approach enabled us to quantify the impact of vaccination with high precision. Furthermore, by analyzing hospitalization and mortality trends across multiple age groups, we have provided a more granular understanding of how vaccination influenced disease severity in different demographic subpopulations within Spain. Such detailed insights have not been previously reported in similar population-based studies.

Vaccination conferred sufficient protection against severe disease and altered the course of the COVID-19 pandemic. Given the conditions of the pandemic, measuring the impact of vaccination directly by comparing a vaccination scenario with a non-vaccination scenario was not possible at a nationwide level. This is why mathematical models are useful for estimating non-vaccination scenarios to achieve such comparisons. Thanks to our estimated scenarios, we could assess the impact of vaccination in Spain. Our approaches generated estimations of hospitalizations and deaths averted as a result of vaccination against SARS-CoV-2.

### 4.4. Limitations

We estimated how the pandemic would have evolved if no vaccines had been available by estimating a new scenario, but we did not include non-pharmaceutical interventions, viral variants, or limitations on mobility that could have altered the viral evolution in the absence of vaccination. It is of interest to mention that the last waves of 2021 in Spain, which were primarily caused by the omicron variant and its descendants (B.1.1.529), presented different characteristics than the previous waves [21,33], but its impact was not included in our model. In addition, forecasting using the time-series signature can be very accurate, particularly when time-based patterns are present in the underlying data. As with most uses of machine learning, the prediction is only as good as the patterns in the data. Forecasting using this approach may not be suitable when patterns are not present or when the future is highly uncertain (i.e., past results are not a suitable predictor of future performance). We could not use ARIMA or MCMC to create the estimated scenario, so we did not compare different approaches. Although it has been found that mortality due to COVID-19 may have been under-reported [41], in Spain, almost all deaths occurred in hospital, so our data can be considered reliable. This is key when modeling and fitting a machine learning algorithm because the final model can only be as good as the provided data. The wide confidence intervals, particularly for the ElasticNet model, reflect the inherent uncertainty in modeling complex phenomena such as pandemic outcomes. This uncertainty arises from potential changes in transmission dynamics, population behavior, and viral variants. While the confidence intervals indicate variability, the consistency in trend direction across models (ElasticNet and RandomForest) suggests that the main conclusions remain robust despite this uncertainty.

## 5. Conclusions

We fit mathematical models to estimate both hospitalizations and deaths due to COVID-19 in a non-vaccination scenario. We determined the impact of vaccination by estimating the hospitalizations and deaths that, otherwise, could have occurred if vaccines had not been administered. In Spain, demographic and clinical profiles shifted significantly during the first months of the pandemic, reflecting the differential impact of early vaccination efforts. Vaccination altered the evolution of the COVID-19 pandemic and prevented up to 24,546 deaths in Spain in 2021. Vaccination reduced not only mortality but also the number of hospitalizations and the burden of the pandemic. Its protective effect was observable shortly after the beginning of vaccination for each age group. Machine learning approaches can be useful in uncertain contexts because a time-series signature can provide accurate forecasts. By integrating machine learning models and age-stratified analyses, our study provides a comprehensive view of the pandemic’s evolution in Spain, demonstrating how targeted vaccination strategies can alter disease trajectories at a national level.

## Figures and Tables

**Figure 1 jcm-13-05890-f001:**
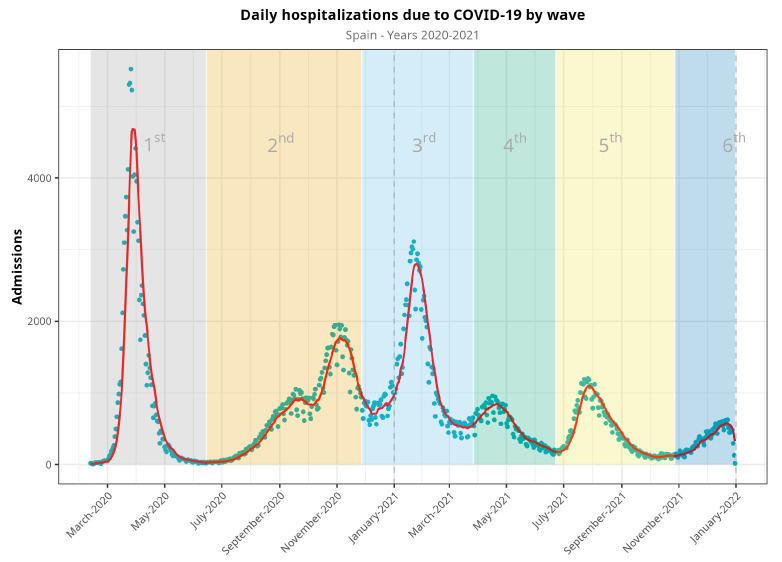
Evolution of hospitalizations during the COVID-19 pandemic in Spain from March 2020 to December 2021, split into waves. Blue dots represent raw data, whereas the red line represents a 7-day moving average of the time series. Vertical dash line at the end of the figure denotes that there were no available data for the sixth epidemic wave (it would continue, otherwise).

**Figure 2 jcm-13-05890-f002:**
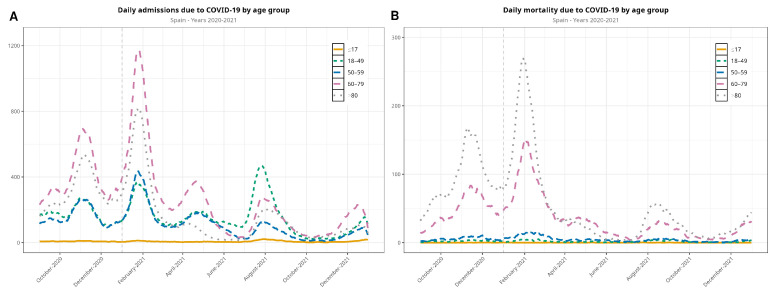
Evolution of the COVID-19 pandemic in terms of hospitalizations (**A**) and in-hospital deaths (**B**) between September 2020 and December 2021 (the first wave is not included in the visualization), disaggregated by age group. The vertical dashed line denotes the new year.

**Figure 3 jcm-13-05890-f003:**
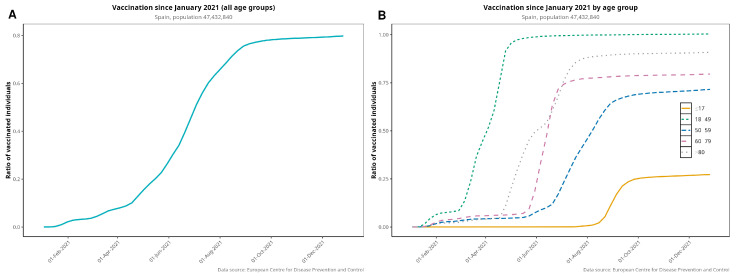
Vaccination rollout in Spain for the entire population (i.e., fully vaccinated individuals (**A**)) disaggregated by age group (**B**). Elderly patients were prioritized to receive the first dose of vaccine.

**Figure 4 jcm-13-05890-f004:**
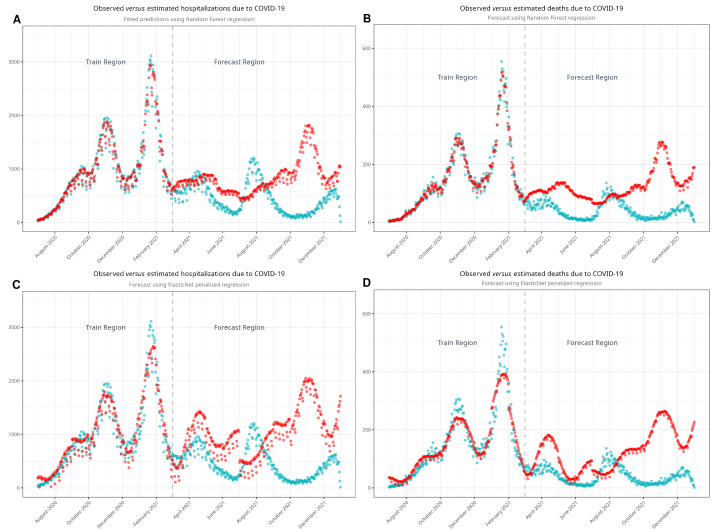
Models developed with random forest (**A**,**B**) and with ElasticNet (**C**,**D**) to estimate non-vaccination scenarios. Turquoise dots represent the observed values, while red dots represent estimated values. Note the good fit of the model in the train region before it estimates the values in the forecast region.

**Figure 5 jcm-13-05890-f005:**
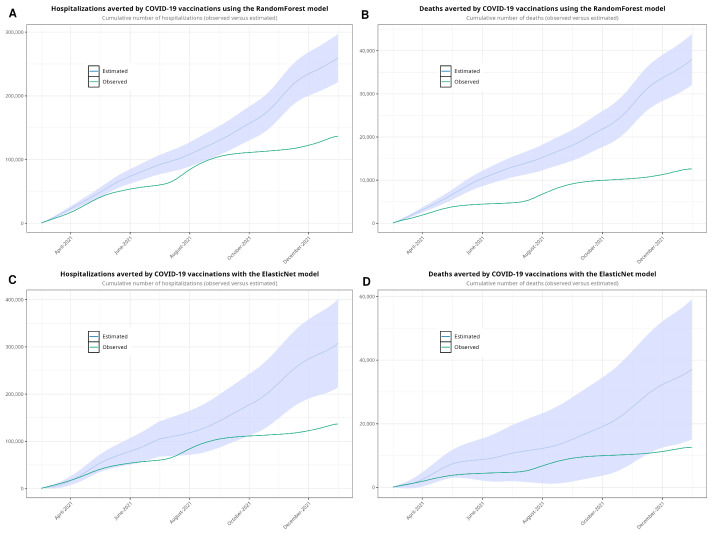
Cumulative sum estimated with random forest (**A**,**B**), showing the observed values and the estimates with 95% confidence intervals, and with ElasticNet (**C**,**D**). We forecasted values from 1 March to 31 December 2021.

**Table 1 jcm-13-05890-t001:** Epidemiological and demographic characteristics of patients hospitalized in Spain between 2020 and 2021.

	All Waves	First	Second	Third	Fourth	Fifth	Sixth	*p* Value
Hospital admissions	498,789	115,356	127,114	126,623	51,006	54,570	24,120	0.001
Sex (men)	56.1%	56.2%	55.5%	56.2%	57.5%	55.3%	56.2%	0.001
Age, median (IQR)	66 (28)	69 (25)	68 (28)	69 (25)	59 (24)	57 (38)	65 (28)	0.001
Hospital stay, median (IQR)	8 (9)	11.9 (9)	8 (9)	12.1 (9)	7 (7)	10.4 (8)	6 (7)	0.001
ICU								
Admissions	54,354	10,218	13,302	14,441	7194	6941	2258	0.001
ICU (%)	10.9	8.9	10.5	11.4	14.1	12.7	9.4	0.001
ICU stay, median (IQR)	10 (21)	11 (22.3)	10 (21.4)	11 (21.5)	11 (21.4)	10 (18.7)	6 (9.1)	0.001
Mortality								
Deaths	71,437	21,037	18,229	20,400	3793	5487	2491	0.001
Mortality rate (%)	14.3	18.2	14.3	16.1	7.4	10.1	10.3	0.001
Comorbidities								
Type 2 diabetes	21.7	20.6	23.2	24.3	18.3	18.3	21.4	0.001
Hypertension	33.8	35.2	34.7	37.2	31.1	25.0	31.6	0.001
AMI	7.1	7.3	7.4	7.9	5.3	5.7	7.5	0.001
CHF	7.2	6.5	7.8	8.4	4.2	6.8	7.9	0.001
Dementia	4.8	5.5	5.4	5.2	2.1	3.9	3.6	0.001
Kidney disease	10.5	10.3	11.2	11.7	6.5	9.7	11.1	0.001
Liver disease	0.4	0.4	0.4	0.5	0.3	0.4	0.5	0.001
Malignancy	5.6	5.4	5.8	6.0	4.2	5.3	7.4	0.001
Obesity	12.9	9.2	12.6	14.0	17.0	14.4	13.8	0.001
COPD	7.4	7.6	7.4	7.9	5.9	6.6	8.8	0.001
CEVD	0.7	0.7	0.8	0.9	0.5	0.6	0.8	0.001

ICU: intensive care unit; AMI: acute myocardial infarction; CHF: congestive heart failure; CEVD: cerebrovascular disease; COPD: chronic obstructive pulmonary disease; IQR: interquartile range. Age is expressed in years. Hospital/ICU stay is expressed in days.

**Table 2 jcm-13-05890-t002:** Outcomes in terms of total admissions, ICU admissions, and mortality of hospitalized patients disaggregated by age group.

	Total	First Wave		Second Wave		Third Wave		Fourth Wave		Fifth Wave		Sixth Wave		*p*-Value
**Admissions**														
≤17	6568	583	(8.9%)	1622	(24.7%)	1021	(15.5%)	657	(10%)	1764	(26.9%)	921	(14%)	0.001
18–49	99,570	18,223	(18.3%)	23,525	(23.6%)	18,542	(18.6%)	13,669	(13.7%)	20,660	(20.7%)	4951	(5%)	0.001
50–59	78,535	18,524	(23.6%)	19,534	(24.9%)	19,432	(24.7%)	11,560	(14.7%)	6262	(8%)	3223	(4.1%)	0.001
60–79	178,667	45,100	(25.2%)	44,052	(24.7%)	48,808	(27.3%)	18,198	(10.2%)	13,267	(7.4%)	9242	(5.2%)	0.001
≥80	125,834	30,642	(24.4%)	35,998	(28.6%)	36,365	(28.9%)	5806	(4.6%)	11,707	(9.3%)	5316	(4.2%)	0.001
**ICU**														
≤17	490	95	(19.4%)	109	(22.2%)	74	(15.1%)	62	(12.7%)	111	(22.7%)	39	(8%)	0.001
18–49	10,227	1493	(14.6%)	2186	(21.4%)	1,957	(19.1%)	1513	(14.8%)	2633	(25.7%)	445	(4,4%)	0.001
50–59	11,187	2117	(18.9%)	2733	(24.4%)	2994	(26.8%)	1705	(15.2%)	1240	(11.1%)	398	(3.6%)	0.001
60–79	28,548	5862	(20.5%)	7159	(25.1%)	8300	(29.1%)	3537	(12.4%)	2526	(8.8%)	1164	(4.1%)	0.001
≥80	2372	342	(14.4%)	754	(31.8%)	687	(29%)	185	(7.8%)	248	(10.5%)	156	(6.6%)	0.001
**Deaths**														
≤l7	27	4	(14.8%)	7	(25.9%)	6	(22.2%)	2	(7.4%)	6	(22.2%)	2	(7.4%)	0.001
18–49	1282	345	(26.9%)	300	(23.4%)	253	(19.7%)	92	(7.2%)	222	(17.3%)	70	(5.5%)	0.001
59–59	3274	905	(27.6%)	760	(23.2%)	861	(26.3%)	277	(8.5%)	343	(10.5%)	128	(3.9%)	0.001
60–79	25,427	7937	(31.2%)	5,914	(23.3%)	7074	(27.8%)	1871	(7.4%)	1733	(6.8%)	898	(3.5%)	0.001
≥80	40,855	11,707	(28.7%)	11,111	(27.2%)	12,041	(29.5%)	1499	(3.7%)	3127	(7.7%)	1370	(3.4%)	0.001

Mortality ratios were calculated by dividing the the presented events in a given age group in a wave by the total population in each group.

**Table 3 jcm-13-05890-t003:** Observed, estimated, and averted events in the first year of vaccination.

	Hospitalizations		Deaths	
	Events	(95% CI)	Events	(95% CI)
**RandomForest**				
Observed	136,658	(NA)	12,595	(NA)
Estimated	251,830	(216,99–286,663)	37,673	(317,13–43,633)
Averted	115,172	(80,339–150,005)	25,078	(191,18–31,038)
**ElasticNet**				
Observed	136,658	(NA)	12,595	(NA)
Estimated	307,617	(214,502–400,733)	37,141	(15,143–59,138)
Averted	170,959	(77,844–264,075)	24,546	(2,548–46,543)

Estimation computed for the period between March, 2021 and December, 2021. We estimated data using the following two models: ElasticNet and random forest. NA: non-applicable; CI: confidence interval.

## Data Availability

A contract signed with the Spanish Health Ministry, which provided the dataset, prohibits the authors from providing their data to any other researcher. Furthermore, the authors must destroy the database upon the conclusion of their investigation. The database cannot be uploaded to any public repository.

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
