# Peer review of "Impact and Effectiveness of COVID-19 Vaccines Based on Machine Learning Analysis of a Time Series: A Population-Based Study"

_jcm, 2024, doi:10.3390/jcm13195890_

Round 1

Reviewer 1 Report

Comments and Suggestions for Authors

Besides the use of real data from Spain, my my concern with this manuscript is what do we learn from this that we did not know before? The conclusion is really not appealing as this is now new "Demographic and clinical profiles changed over the first months of the pandemic. Patients over 80 years old and other age groups obtained clinical benefit from early vaccination. The severity of COVID-19, in terms of hospitalizations and deaths, decreased due to vaccination." This should not be generalized, but Spain specific "Demographic and clinical profiles IN SPAIN changed over the first months of the pandemic. Patients over 80 years old and other age groups obtained clinical benefit from early vaccination WORLDWIDE???. The severity of COVID-19, in terms of hospitalizations and deaths IN SPAIND, decreased due to vaccination."

Author Response

Dear Reviewer,

Thank you for your detailed feedback and for highlighting areas of potential improvement in our manuscript. We understand your concern regarding the novelty and generalizability of our findings.

The reviewer’s primary concern revolves around novelty and generalizability, and feels that the conclusions drawn in our manuscript are not particularly new or groundbreaking. Specifically, the points that the demographic and clinical profiles changed and that vaccination decreased the severity of COVID-19 are already well-established facts worldwide. The reviewer is questioning what our study adds to the existing literature.

We agree that our conclusions should be more specific to Spain, given that our study used real-world data from our country.

In addition, the reviewer is concerned with how the findings are framed. The reviewer suggests that our conclusions might be too broad and not adequately tailored to the specific context of Spain. For example:

  • The phrase "Patients over 80 years old and other age groups obtained clinical benefit from early vaccination" might suggest that this is a global conclusion, while your data are based on Spain specifically.

  • Similarly, when saying "The severity of COVID-19, in terms of hospitalizations and deaths, decreased due to vaccination," it should be made clear that this observation applies to Spain, based on the data from your study.

The key issue is that our findings should be contextualized within Spain, as the study uses real-world data from Spain. The reviewer would like us to be explicit that our conclusions are derived from this specific population rather than generalizing globally.

We have revised the manuscript to emphasize that our findings are specific to the Spanish context. We believe these adjustments will better reflect the contribution of our study.

Additionally, we have enhanced the discussion to specify how our study adds to the existing literature by highlighting unique aspects of Spain's vaccination strategy and the impact of early vaccination in the elderly population.

After having carefully considered your feedback, we made the following modifications to the manuscript (please, see the highlighted sentences in yellow in the manuscript):

  1. Abstract: We revised the abstract to specify that our findings are based on data from Spain and to emphasize the unique aspects of our study, such as the use of machine learning models for estimating averted hospitalizations and deaths.

  2. Introduction and Discussion: We included additional statements to contextualize our results as Spain-specific and to highlight the unique contributions of our study, including the high level of demographic granularity and the detailed modeling of vaccination scenarios.

  3. Conclusions: We reframed the conclusions to ensure they are specific to Spain and added a statement to reinforce the novel contributions of our research.

Specifically, we have done the following changes:

Abstract

We explicitly mentioned the Spain-specific context and highlight what makes this study unique.

Original:
“Patients over 80 years old and other age groups obtained clinical benefit from early vaccination. The severity of COVID-19, in terms of hospitalizations and deaths, decreased due to vaccination.”

Revised:
In Spain, patients over 80 years old and other age groups experienced significant clinical benefits from early vaccination, which contributed to a marked reduction in COVID-19-related hospitalizations and deaths. Our use of machine learning models provided a detailed estimation of the averted burden of the pandemic, demonstrating the effectiveness of vaccination at a population-wide level.

Introduction

We added a short statement to emphasize the rationale of why studying the Spanish population specifically adds value.

Given the unique characteristics of the Spanish healthcare system and the country’s age-stratified vaccination strategy, studying Spain offers an opportunity to understand the differential impact of vaccination across diverse demographic groups, contributing insights that are not directly generalizable to other populations.

Discussion

Here, we addressed the reviewer’s concern about the generalizability of our findings and clearly articulate what your study adds compared to previous research. We emphasized Spain-specific context by modifying the statements that sound too general to ensure they reflect your country-specific findings.

Original: “The severity of COVID-19 decreased due to vaccination in all age groups.”

Revised:In Spain, vaccination led to a significant reduction in the severity of COVID-19 across all age groups, with particularly marked benefits observed in the elderly population.

We also highlighted the novelty of our study by adding a paragraph detailing what our study brings to the table in terms of the use of machine learning and the high level of granularity in demographic analysis. We added:
This study stands out due to the use of advanced machine learning algorithms, such as ElasticNet and RandomForest, which allowed us to create accurate non-vaccination scenarios. This approach enabled us to quantify the impact of vaccination with high precision. Furthermore, by analyzing hospitalization and mortality trends across multiple age groups, we have provided a more granular understanding of how vaccination influenced disease severity in different demographic subpopulations within Spain. Such detailed insights have not been previously reported in similar population-based studies.”

We also put our results into context by adding a brief comparison with global trends and clarifying that the patterns observed in Spain may differ in other settings. We added:
While the overall reduction in hospitalizations and deaths due to vaccination is consistent with global findings, the timing and magnitude of these changes in Spain were influenced by the country’s specific vaccination rollout strategy and healthcare infrastructure.

Conclusion

We have reframed the conclusion to reinforce the Spain-specific findings and highlight the study’s unique contributions.

Original: “Demographic and clinical profiles changed over the first months of the pandemic.”

Revised:In Spain, demographic and clinical profiles shifted significantly during the first months of the pandemic, reflecting the differential impact of early vaccination efforts.

Finally, we added a novelty statement by including a final sentence that reiterates the novel contribution of your study:
By integrating machine learning models and age-stratified analyses, our study provides a comprehensive view of the pandemic’s evolution in Spain, demonstrating how targeted vaccination strategies can alter disease trajectories at a national level.

We believe these changes address your concerns and improve the clarity and impact of our manuscript. Thank you again for your valuable suggestions.

Kind regards,

Rafael Garcia-Carretero, MD, PhD

Reviewer 2 Report

Comments and Suggestions for Authors

Dear authors,

I appreciate the opportunity to review your interesting paper. Please see my comments below:

- Page 3, line 107: Why were patients with unknown length of stay excluded from the study?

- Page 3, line 109, “We categorized the pandemic following a previous epidemiological study [21]”. The cited reference is not accessible. Please correct the reference since it is a fundamentally important part of the study methodology.

- Please summarize the Materials and Methods section. Most parts of the Materials and Methods section (such as lines 120-127 on page 3, whole section 2.5, and most parts of sections 2.6-2.7) should be removed or transferred to the Introduction since they are unrelated to this study’s methodology.

- Figures 2 and 3: A different set of colors is recommended to make the lines more distinguishable.

- The confidence intervals for the estimated hospitalizations and deaths are quite wide, particularly for the ElasticNet model. While this reflects uncertainty in the model, the manuscript would benefit from a discussion of how this uncertainty might affect the robustness of the conclusions.

- The authors are strongly recommended to discuss previous similar studies (whether in Spain or not) in the discussion, and compare the results.

- Please reconsider using the word “global” (lines 12, 17, 237,…) since the data are limited to Europe at best.

- The manuscript needs serious language editing.

Comments on the Quality of English Language

- The manuscript needs another round of language editing. Some redundant sentences should be removed.

Author Response

Dear Reviewer,

Thank you for your detailed feedback and for highlighting areas of potential improvement in our manuscript.

Comment #1.- Page 3, line 107: Why were patients with unknown length of stay excluded from the study?

We are aware that the reviewer wants to know why patients with incomplete data (specifically, unknown length of stay) were excluded. We have added a brief explanation justifying why excluding these patients was necessary, as follows:

We excluded patients with unknown length of stay to ensure the accuracy and completeness of the dataset. As length of stay is a key outcome variable in the analysis of disease severity and healthcare utilization, including patients with missing values could bias the results and reduce the robustness of our conclusions.

Comment #2.- Page 3, line 109: “We categorized the pandemic following a previous epidemiological study [21]”. The cited reference is not accessible. Please correct the reference since it is a fundamentally important part of the study methodology.

We are aware that the reviewer could not access this reference. Since it is a key element in defining the waves, it’s essential to either provide a valid citation or describe the method in more detail. The previous reference was a URL from the Spanish Government. However, since we began to write this research, new papers have been published that made a epidemiological study on Spanish population. We updated the reference with a new one. We apologize for the inconvenience. The new publication is: https://doi.org/10.1186/s12879-023-08454-y.

Comment #3.- Please summarize the Materials and Methods section. Most parts of the Materials and Methods section (such as lines 120-127 on page 3, whole section 2.5, and most parts of sections 2.6-2.7) should be removed or transferred to the Introduction since they are unrelated to this study’s methodology.

We are aware that the reviewer feels that the Methods section is too detailed and that some content should be moved to the Introduction. We have summarized the detailed methodology, focusing only on the key points, and move any context-setting information to the Appendix or Supplementary Data. Here’s our approach:

  • Section 2.5: Estimated Scenarios of the Unvaccinated Population: We have shorten by focusing on how the models were applied rather than justifying the choice.

    • Revised Version: “We developed a population-based, epidemiological study to compare two scenarios: observed hospitalizations and deaths before and after vaccination, and an estimated non-vaccination scenario using time series and machine learning models.

  • Section 2.6: Mathematical Modeling of Hospitalizations and Mortality: We have condensed to key points.

    • Revised Version: “We utilized ElasticNet and RandomForest models to forecast the impact of vaccination by fitting the models to a training dataset from July 2020 to February 2021 and validated them using a testing set. Each model was evaluated using cross-validation metrics.

  • Section 2.7: Machine Learning Algorithms: We also condensed detailed technical explanations to the appendix/supplementary material.

    • Revised Version: “The models included ElasticNet for linear predictions and RandomForest for capturing non-linear patterns. Each algorithm was tuned to achieve optimal performance.

Comment #4.- Figures 2 and 3: A different set of colors is recommended to make the lines more distinguishable.

We are aware that the colors in Figures 2 and 3 may not be visually distinct. We updated the color scheme to improve clarity. We used a colorblind-friendly palette to ensure accessibility, and we added different line styles (e.g., dashed, dotted) to further differentiate the curves.

Comment #5.- The confidence intervals for the estimated hospitalizations and deaths are quite wide, particularly for the ElasticNet model. While this reflects uncertainty in the model, the manuscript would benefit from a discussion of how this uncertainty might affect the robustness of the conclusions.

Regarding wide confidence intervals, we addressed this concern in the Discussion section. We added a paragraph in the Limitation section addressing this concern as follows:

The wide confidence intervals, particularly for the ElasticNet model, reflect the inherent uncertainty in modeling complex phenomena such as pandemic outcomes. This uncertainty arises from potential changes in transmission dynamics, population behavior, and viral variants. While the confidence intervals indicate variability, the consistency in trend direction across models (ElasticNet and RandomForest) suggests that the main conclusions remain robust despite this uncertainty.

Comment #6.- The authors are strongly recommended to discuss previous similar studies (whether in Spain or not) in the discussion, and compare the results.

We have included a paragraph that compares our findings with those of other studies, either in Spain or internationally. As follows:

Previous studies conducted in Spain (e.g., Barandalla et al., 2021) also reported a significant reduction in hospitalizations following the vaccination rollout, using different modeling approaches. Our study adds to these findings by incorporating machine learning methods and providing a more granular age-stratified analysis, which is lacking in other reports. International studies, such as Watson et al. (2022), have demonstrated similar impacts of vaccination on a global scale, supporting the observed trends in our Spanish cohort.

Comment #7.- Please reconsider using the word “global” (lines 12, 17, 237,…) since the data are limited to Europe at best.

We are aware that using “global” may give a misleading impression since your data are specific to Spain and, at best, Europe. We have replaced “global” with “nationwide” or “Spain-specific” where applicable. For instance, we replaced “global reduction of hospitalizations” with “nationwide reduction in hospitalizations observed in Spain.

Comment #8.- The manuscript needs serious language editing. The manuscript needs another round of language editing. Some redundant sentences should be removed.

We apologize as the reviewer feels that the language could be improved for better readability and conciseness.

We thank the reviewer for your feedback regarding the language quality of our manuscript. We would like to clarify that the manuscript was professionally edited by International Edit, a reputable scientific editing service dedicated to translating and refining academic manuscripts for publication. We opted for this service to ensure that the language meets the high standards required for international scientific publications.

We believe that the current language quality is suitable for submission, as the manuscript has undergone a thorough review by experienced editors specializing in academic English. However, if there are specific sections that you feel require improvement, we are happy to review them and make the necessary adjustments.

Anyway, we conducted a thorough language review, focusing on removing redundant sentences, using clear, precise language, and ensuring consistent terminology.

We believe these changes address your concerns and improve the clarity and impact of our manuscript. Thank you again for your valuable suggestions.

Kind regards,

Rafael Garcia-Carretero, MD, PhD

Round 2

Reviewer 2 Report

Comments and Suggestions for Authors

I appreciate the authors’ efforts in revising the manuscript considering the comments and suggestions.